# Therapeutic Transplantation of Human Central Nervous System Organoids for Neural Reconstruction

**DOI:** 10.3390/ijms25158540

**Published:** 2024-08-05

**Authors:** Sung Jun Hong, Minsung Bock, Songzi Zhang, Seong Bae An, Inbo Han

**Affiliations:** 1Research Competency Milestones Program (RECOMP), School of Medicine, CHA University, Seongnam-si 13488, Republic of Korea; sjhong1021@gmail.com; 2Department of Medicine, School of Medicine, CHA University, Seongnam-si 13496, Republic of Korea; 3Department of Neurosurgery, CHA Bundang Medical Center, CHA University, Seongnam-si 13496, Republic of Korea; minsungbock@gmail.com (M.B.); szzhang95@gmail.com (S.Z.); ansb31@chamc.co.kr (S.B.A.)

**Keywords:** central nervous system, organoid, neural reconstruction, transplantation

## Abstract

Damage to the central nervous system (CNS) often leads to irreversible neurological deficits, and there are currently few effective treatments available. However, recent advancements in regenerative medicine have identified CNS organoids as promising therapeutic options for addressing CNS injuries. These organoids, composed of various neurons and supporting cells, have shown potential for direct repair at injury sites. CNS organoids resemble the structure and function of actual brain tissue, which allows them to adapt and function well within the physiological environment when transplanted into injury sites. Research findings suggest that CNS organoids can replace damaged neurons, form new neural connections, and promote neural recovery. This review highlights the emerging benefits, evaluates preclinical transplantation outcomes, and explores future strategies for optimizing neuroregeneration using CNS organoids. With continued research and technological advancements, these organoids could provide new hope for patients suffering from neurological deficits.

## 1. Introduction

The landscape of neurological research and therapy faces immense challenges due to the complexity of central nervous system (CNS) diseases, which often result in significant morbidity and remain largely incurable. These CNS diseases not only impact the individuals but also cause tremendous social cost. For instance, about 55 million people worldwide suffer from traumatic brain injury (TBI) and for this over 400 billion USD per year is spent to deal with the injury itself and associated disability [1]. In case of spinal cord injury (SCI), 23 million people per year suffer from the injury and struggle with life-long disability [2]. Another global statistics in 2016 demonstrated that neurological disorders in general, including Parkinson’s and Alzheimer’s disease, are mainly responsible for 276 million disability-adjusted life years and 9 million deaths per year [3]. Likewise, most CNS disorders lack effective long-term treatments, affecting numerous people worldwide who are desperately waiting to be cured. 

Traditionally, most CNS diseases were treated by physical or chemical interventions: TBI with hematoma removal, PD with chemicals like L-DOPA or deep brains stimulation, and SCI with decompression and stabilization. Even though these approaches are clinically acknowledged to alleviate the symptoms, the traditional treatment modalities have struggled to replicate the intricate neural architectures and physiological responses of the human CNS, leading to a substantial gap in effective therapeutic interventions and comprehensive disease understanding. Moreover, drug development for these conditions often faces poor outcomes due to inadequate disease models [4]. In this context of urgent and unmet needs, therapeutic materials for the CNS diseases have been developing rapidly. At an early stage, biocompatible materials such as collagen and hydrogel were applied to stimulate internal neural regeneration of the injured site. As stem cell technology evolved, researchers started to utilize stem cells to treat CNS diseases, which was not so successful due to the low cell survival and differentiation efficiency [5]. This could be partly solved by loading them on the biocompatible materials, used as scaffolds and exosomes, so that neural reconstruction can firmly be induced by transplanted cell source [6,7]. These stem cell-based therapeutics were spotlighted as an effective approach for curing direct injuries of brain and spinal cord diseases owing to their own advantages such as direct cell differentiation and tunable scaffolds [8,9,10,11]. Yet, the survival rate and differentiation potential still lacked and other problems such as low transplant-host interaction are left as hurdles [12].

The advent of organoid technology, particularly CNS organoids, represents a transformative advancement. These three-dimensional (3D), miniaturized, and simplified versions of the brain or spinal cord offer a promising platform for understanding neurological diseases and advancing regenerative therapies. Recent research has highlighted the significant potential of CNS organoids in disease modeling, drug screening, and cellular therapy and partial replacement of animal models which have been arousing ethical issues [13]. CNS organoids are derived from pluripotent stem cells that can be induced to develop into complex structures resembling various regions of the human brain or spinal cord [14,15]. The versatility of these organoids lies in their ability to mimic the physiological and pathological conditions of the CNS, offering insights into the intricacies of neurodevelopment and disease pathogenesis [16]. For instance, CNS organoids have been utilized to model Alzheimer’s disease, successfully replicating hallmark features such as β-amyloid aggregation and tauopathy, which were previously difficult to achieve in two-dimensional (2D) cultures [17,18].

As the fundamentals of CNS organoids become more established, their applications for clinical use are being extensively discussed. CNS organoids provide a unique platform for drug testing, reducing reliance on animal models, which are often inaccurate for humans and pose ethical issues [16]. Notably, about 80% of preclinical drugs screened using animal models fail at the translational stage to humans, and over 110 million mice are sacrificed in this process annually in the USA [19]. The paradigm shift from animal models to organoids is considered a potential breakthrough in the field of drug screening. For instance, research using CNS organoids in a model of Parkinson’s disease to screen for neuroprotective agents has identified several compounds that mitigated dopaminergic neuron loss, demonstrating the organoids’ capability as effective drug testing platforms [20]. These findings are significant as they not only accelerate the discovery process, but also improve the specificity and efficacy of therapeutic agents tailored to individual patients.

The therapeutic potential of CNS organoids extends to direct applications in regenerative medicine for irremediable neural diseases [21,22,23]. Transplantation of organoids into damaged areas of the rodent brain has demonstrated integration with host tissues and partial restoration of function, suggesting the potential for similar strategies in humans. Groundbreaking studies have reported that transplanted brain or spinal cord organoids could bridge injury sites and support the reformation of functional neuronal networks in models of traumatic brain injury and spinal cord injury. The brain and spinal cord tissue-like features of CNS organoids, including neural cell types and extracellular matrix components, are crucial for the effective regeneration of the injured tissue, repair of neural networks, and rehabilitation of neural functions [24]. Future strategies to improve the therapeutic efficacy of CNS organoids include optimizing organoid vascularization and maturation to ensure better integration and functionality after transplantation [25]. Some researchers have proposed that incorporating microfluidic systems to simulate blood flow within organoids could promote the development of a vascular network, thereby improving nutrient and oxygen supply and more closely mimicking the in vivo environment [26].

CNS organoids represent a significant advancement in neurological research and regenerative medicine. The ability to generate organoid models of the CNS not only enables a deeper understanding of neurodevelopmental and neurodegenerative processes but also holds the promise of revolutionizing therapies for debilitating neurological conditions. As research progresses, the translation of CNS organoid technology from the laboratory to clinical applications appears increasingly feasible, opening new avenues for treating conditions once considered untreatable. In this review article, we present the key characteristics of CNS organoids harnessed for neural repair and discuss recent preclinical applications of CNS organoid in therapeutic contexts, drug testing, and transplantations. Combinations of relevant keywords—namely, “CNS (brain or spinal cord) organoids,” “CNS injury,” “CNS diseases,” “neural regeneration,” “drug screening,” “transplantation,” “engraftment”—were used to search articles on PubMed (https://pubmed.ncbi.nlm.nih.gov/ (accessed on 2 June 2024)) and Google Scholar, published mainly from January 2013 to May 2024.

## 2. CNS Organoids as Therapeutics

Recovery from neural injuries requires a series of neurogenic processes, including vascularization, neural differentiation, synapse formation, and axonal growth. In this context, CNS organoids hold great potential as neuroregenerative treatments for various diseases due to their unique characteristics that stem cells cannot provide. CNS organoids are composed of diverse cell types, including neural progenitors and other cells that have already differentiated in vitro [27,28,29,30]. This composition is highly advantageous for regenerative medicine because the core purpose of the therapy is to reestablish the injured tissue as a functional whole, not just a specific type of cells. A transplanted cerebral organoid contains numerous neural progenitors, which can readily migrate and differentiate into appropriate cells in response to signaling molecules from the surrounding environment. Differentiated neurons extend their axons guided by growth factors and the extracellular matrix within the engrafted organoid and the nearby host tissue [31]. This axonal growth leads to formation of synapses (synaptogenesis) between newly differentiated neurons, reconnecting neural circuits between the injured site and the host tissue [25].

Stem cell-based neurotherapy can promote differentiation, axonal growth, and synaptogenesis to some extent, but it often fails to induce vascularization [32,33]. In contrast, the vascularization of CNS organoids has been successfully reported both in vitro and in vivo [25,34,35,36]. This vascularization is considered a crucial factor determining the therapeutic performance of cerebral organoids. In contrast to other organs, the brain facilitates the successful transplantation of organoids with a sufficient vascular system, allowing easy infiltration into the engrafted organoid with no need for mesenchymal and endothelial cells [25]. This leads to a higher survival rate, providing opportunities for cells to be fully differentiated, matured, and integrated into the host CNS [37].

Moreover, the cellular compositions of CNS organoids can closely resemble CNS tissues in vivo [38] and can be controlled by extrinsic and intrinsic morphogenetic factors as illustrated in Figure 1 [39,40,41]. These characteristics ensure high tissue compatibility compared to neural stem cells or cerebral organoids in general, as most features of the target tissues can be pre-adjusted in vitro. Region-specific CNS organoids (e.g., midbrain [22,42,43], cerebellum [44,45,46], and spinal cord [47,48,49]) can promote the recovery of injured tissue, as they already contain cells specific to the target regions within CNS. In practice, Daviaud et al. [32] directly demonstrated the superiority of CNS organoid transplantation over neural stem cell transplantation. The researchers confirmed that cerebral organoid grafts showed better therapeutic effects, as reflected by higher survival rates and improved neurodifferentiation. The microenvironments, tissue compatibility, and cellular homogeneity of the transplanted organoids are key factors contributing to their superior capability for neural reconstruction. The detailed advantages of CNS organoid transplantation (in cases of traumatic brain injury) over stem cell therapy are summarized in Table 1.

## 3. CNS Organoids Transplantation

### 3.1. Traumatic Brain Injury

Traumatic brain injury (TBI) involves the impairment of typical brain function resulting from a sudden mechanical impact to the head. Despite its increasing incidence and the consequential disability and morbidity it inflicts globally, there remains a significant gap in research aimed at comprehensively understanding and developing therapeutic interventions for this condition. TBI can be categorized into acute and chronic phases. The acute phase occurs immediately after the mechanical impact, leading to axonal damage and vascular dysfunction. Subsequently, in the chronic phase, the brain injury is exacerbated by sustained inflammation, oxidative stress, and cytotoxicity, culminating in neurodegeneration and vascular pathologies [50].

TBI is primarily modeled using rodents with various impact systems (e.g., concussion, impact, rotational, and blast injuries) tailored to the specific research purpose. Among these, the controlled cortical impact (CCI) model **(**Figure 2) is commonly used to study the effects of direct physical impacts on the cortex and postoperative recovery after targeted surgery [53,54]. The CCI model easily provides physical space for transplanting stem cells or organoids, making it advantageous for researchers to conduct longitudinal investigations of the therapeutic effects of the engraftments. Recently, Ramirez et al. [55] attempted to use human brain organoids to bridge the gap between rodent and human neurons and to establish an animal-free TBI model. The TBI model created with cerebral organoids demonstrated major pathological and metabolic characteristics after the impact, sufficiently mimicking rodent models. However, the system has not yet been standardized for widespread use in studying TBI and its treatment.

To investigate the therapeutic effect of brain organoid transplantation, different degrees (mild to severe) of TBI have been modeled. Kim et al. [54] explored the therapeutic potential of cortical organoids in mild TBI, characterized by secondary damage such as neuronal loss, synapse reduction, and dendrite degeneration, despite the preservation of motor function and brain structure. The organoids, generated from hESCs, were grafted into a mild TBI mouse model. The grafted cerebral organoids survived post-transplantation, and cortex tissue was successfully reconstructed. Fluoro-Jade B (FJB) staining revealed a significantly reduced number of FJB^+^ cells, indicating the neuroprotective effect of the cerebral organoids. Vascular-like structures were observed in the cerebral organoids 14 days after implantation. Moreover, immunostaining revealed that most of the transplanted cells differentiated into immature neurons (TUJ1^+^). Cognitive improvements in the injured mice were measured using Novel Object Recognition testing. Bao et al. [53] implanted in vitro hESCs-derived cerebral organoids to examine the therapeutic effects in moderate to severe TBI. The implanted cerebral organoids displayed appropriate growth and progressive differentiation over time in vivo. Importantly, the organoids reduced GFAP expression, improved glial scar formation, and promoted neural repair. The grafted cerebral organoids also exhibited spontaneous action potentials under simulation, suggesting that the grafted organoids could transmit neural information. The Morris water maze test revealed improvements in learning and memory abilities in the mice.

Effective TBI recovery through engraftment requires appropriate vascularization of cortical organoids [56]. Human cerebral organoid transplantation in a TBI mouse model led to successful infiltration of host vasculature on the vasculature-rich surface [25]. The transplanted organoids, integrated with the host vasculature, survived and formed synaptic connectivity and axon projections with host neurons while maintaining their own neuronal activity. Vascularization and cellular organization were more effective with cerebral organoid transplantation than with neural stem cells, as demonstrated by higher cell proliferation and differentiation efficiency in the TBI mouse model with cerebral organoid transplantation [32]. Vasculature can also be generated within human cortical organoids in vitro before transplantation [35]. The cerebral organoids co-cultured with endothelial cells from the human umbilical vein formed a tubular vascular system within the organoids. These vascularized cerebral organoids can be transplanted into damaged brain to regenerate the neural circuit and tracts. 

A study by Kitahara et al. [31] proposed that the age of cerebral organoids is crucial for determining the formation of corticospinal tract. The transplantation of 6-week-old cerebral organoids into a TBI mouse model led to more active axon extension and graft overgrowth than when 10-week-old organoids were transplanted. Another study performed similar experiments with 55- and 85-day-old organoids, demonstrating that 55-day-old organoid implantation showed greater therapeutic effects in terms of neurogenesis, anti-inflammation, and synapse regeneration [57]. These studies highlight the importance of the age of implanted organoids for the recovery of injured lesions. 

Other studies have reported successful engraftment of human cerebral organoids, as shown by axon formation and proper synaptic connections with host neurons [21,58,59]. A study found that the transplantation of human cerebral organoids into the prefrontal cortex partially restored the response to auditory stimuli through newly formed subcortical axon projections [58]. Similarly, neural activity investigated by optogenetic activation of reward-seeking behavior after cortical organoid transplantation showed that the transplant could be well integrated into the neural circuit [21].

### 3.2. Spinal Cord Injury

Spinal cord injury (SCI) occurs when the spinal cord is damaged, often resulting from trauma such as accidents or falls. The clinical outcomes for patients with SCI vary widely depending on the severity and location of the injury. In general, SCI can lead to partial or complete loss of sensation, movement, and control below the level of injury, which can be long-lasting. Patients may experience paralysis, loss of bowel and bladder function, respiratory problems, and chronic pain. Rehabilitation efforts focus on maximizing remaining function, managing complications, and improving quality of life for individuals living with SCI [60,61]. There are various types of animal models for SCI (e.g., contusion, compression, distraction, hemisection, and transection), each chosen based on the study’s objectives [62]. Among them, thoracic complete transection model (Figure 2) is a popular choice for organoid transplantation due to the physical space it provides for engraftment and the straightforward tracking of the recovery process [23,34,49,63]. 

For SCI treatment, neural progenitor cell (NPC) transplantation, often combined with bioengineered scaffolds, has been rigorously studied [8,64,65,66,67,68]. Most studies have demonstrated successful therapeutic effects of NPC transplantation, albeit with varying degrees of spinal cord repair. This raises expectations for the therapeutic potential of spinal cord organoid transplantation for SCI. However, organoid transplantation for SCI has been less studied because the fabrication process of spinal cord organoids itself is not yet clearly standardized. 

Despite the nascent stage of spinal cord organoid application for SCI, some groups are exploring its potential. Wang et al. [63] directly compared the effectiveness of stem cells, various scaffold types and morphologies, and cerebral organoids in treating SCI models, quantifying recovery using the Basso, Beattie, and Breshnahan (BBB) locomotor rating scale. The implanted cerebral organoid within Matrigel fused with certain cells, such as neural stem cells, astrocytes, and superficial cortical cells, but not with proliferating radial glial cells or the deep-layer cortical neurons. Significant axon regeneration was observed, suggesting that the local environment is critical for the survival of specific neurons. Intriguingly, the implanted cerebral organoids fused with the recipient tissue on the ventral side, remaining as neural stem cells, astrocytes, and superficial cortical neurons but not as proliferating radial glial cells or deep-layer cortical neurons. The cerebral organoid engraftment group exhibited the highest density of activated microglia. Immunostaining for Iba1^+^ cells showed that Matrigel-based implant and cerebral organoid groups did not differ significantly in axonal regeneration and locomotor function recovery. Xu et al. [49] generated spinal cord organoids by reprogramming human astrocytes and activating FGF, sonic hedgehog (SHH) and bone morphogenetic protein (BMP) signaling. By regulating the OCT4 and p53 genes of hiPSCs along with a small molecule cocktail (CHIR99021, SB431542, RepSox, YH27632, and Op5333-CSBRY), the researchers directly generated human astrocyte-derived organoids (hAD-Organs). These were further patterned into spinal cord organoids by introducing ventralizing and dorsalizing morphogens. The organoid developed spinal cord cellular identity and cytoarchitecture. Although the grafted hAD-Organs survived, differentiated into spinal cord neural cells, formed synapses, and bridged the injured spinal cord tissue, the locomotor functions of SCI mice did not notably return after 6 weeks of transplantation. 

Recently, Sun et al. [23] demonstrated that rodents with SCI recovered after the engraftment of spinal cord motor neuron organoids. The study focused on the types of ECM, which contain various proteins affecting the regenerative potential of spinal cord organoids. Compared to Matrigel and decellularized adult spinal cord ECM (DASCM), decellularized neonatal spinal cord ECM (DNSCM) effectively promoted the proliferation and differentiation of neural progenitor cells in the transplanted spinal cord organoids. Key proteins in DNSCM, such as tenascin and pleiotrophin, were found to advocate axonal growth of engrafted spinal cord organoids. Moreover, the physical properties (e.g., material composition and fiber diameter) of DNSCM were more influential in guiding the early growth of spinal cord organoids and neural progenitor cells, along with exerting an anti-inflammatory effect. Fan et al. [34] transplanted pre-vascularized and linearly oriented human spinal cord-like tissues (VSCT) for treating SCI. Compared to the transplantation of human fetal spinal cord-derived functional cells alone, the addition of human umbilical vein endothelial cells led to higher survival rates, notable neural regeneration, and neurovascular regeneration by promoting the migration of perivascular cells to the transplantation site. During neural recovery, scar formation was significantly suppressed, and the BBB score was the highest in the VSCT-transplanted group, indicating significant neural recovery.

These results highlight the importance of scaffolds in transplanting spinal cord organoids, similar to findings in neural stem cell transplantation [68]. This suggests that researchers should focus not only on the quality of spinal cord organoids, but also on the surrounding microenvironment for the successful regeneration of injured spinal cord tissues.

### 3.3. Parkinson’s Disease (PD)

Parkinson’s disease (PD) is the second most common neurodegenerative disease, characterized by the progressive loss of dopamine-producing neurons in substantia nigra of the brain, caused by factors such as aging and genetics. This neuronal loss results in a range of motor symptoms, including tremors, rigidity, bradykinesia, and postural instability. Additionally, non-motor symptoms such as cognitive impairment, mood disturbances, and autonomic dysfunction can occur, significantly diminishing patients’ quality of life and imposing a substantial and steady economic burden due to the need for continuous treatment without distinct improvement [69].

Unlike TBI and SCI, which are primarily caused by trauma, PD arises from factors including aging and genetics. To elucidate the underlying mechanisms, human midbrain organoid models for PD have been extensively studied in order to understand the disease pathogenesis and facilitate drug discovery [70]. As the etiology of PD varies among patients, PD organoid models are often fabricated from patient-derived cells to investigate personalized disease pathology and treatment. For instance, Kim et al. [71] devised midbrain organoids using iPSCs from PD patients with LRRK2-G2019S mutations, recapitulating the pathological progression seen in patients. Specifically, the PD organoid demonstrated progressive pathology of PD by aggregation of α-synuclein. This LRRK2-G2019S-mutated PD organoid model was employed to screen a mutant-specific drug, an LRRK2-specific-inhibitor, which restored mitochondrial abnormalities and prevented the degeneration of dopaminergic neurons [20]. The pathological hallmarks, increase in α-synuclein and decrease in neuromelanin, could clearly be observed midbrain organoids [72,73]. In patient-derived human midbrain organoid, it was suggested the degree of senescence of astrocytes is associated with pathological α-synuclein which has certain contribution to induction of senescence at beginning stage of PD [74]. These models hold promise for developing personalized drugs for PD with different underlying causes.

Beyond personalized medicine, functional cell transplantation using stereotactic surgery (Figure 2) is considered an effective direct therapy for restoring dopamine levels in PD patients [75]. In 2016, the safety and therapeutic effects of transplanting NPCs from human fetal midbrain were studied using rodent models of PD [76]. As the transplanted progenitor cells originated from midbrain tissue, they primarily differentiated into dopaminergic neurons, which remained at the transplanted site and partially restored impaired motor functions. This led to the first clinical study of human fetal midbrain-derived NPC transplantation in PD patients, which demonstrated both long-term safety and dose-dependent motor function recovery (up to 40%) in humans [77]. 

In addition to progenitor cells, human midbrain organoids are being investigated for PD treatment, although none have been transplanted into patients yet. The transplantation of human pluripotent stem cell-derived midbrain organoids into rodent PD models showed clear advantages over fetal ventral mesencephalon tissue transplantation due to the homogeneous environment of the midbrain organoids relative to the host tissue [43]. hESC-derived organoids were able to retain their ventral midbrain pattern in the NSC culture with midbrain-specific marker expressions. Cultured organoids contained an astrocytic progenitor population that later aided the maturity and function of differentiated midbrain dopaminergic neurons. The cultured cells displayed low levels of apoptosis, senescence, oxidative stress, and mitochondrial stress. Rotation scores were completely recovered up to 9 months after transplantation in all rats. Immunohistochesmistry revealed the engraftments of hGFAP^+^ human astrocytes surrounding TH^+^ cell bodies, with neurite outgrowth from the TH^+^ cell bodies extending toward the host striatum. Whole midbrain organoid engraftment in the rodent PD model not only normalized dopamine levels, but also re-innervated the striatum neural circuit and alleviated motor dysfunction [22]. Although many studies have demonstrated the therapeutic potential of brain organoid transplantation, the appropriateness for brain disorders like PD remains debated, as unavoidable damage to the host brain tissue can occur during the transplantation of the organoids into deep brain regions (Table 2) [43].

## 4. Challenges and Perspectives

CNS organoid transplantation offers unprecedented opportunities to researchers, neurologists, neurosurgeons, and patients with currently incurable CNS diseases, providing therapeutic effects previously unattainable with traditional approaches. For broader application of CNS organoid transplantation, meticulous studies are required on both the organoids themselves and the transplantation methodology. 

Despite advancements, the fabrication and analysis technologies for CNS organoids are not yet fully developed, necessitating further basic research. For instance, the age and size of the engrafted organoid significantly impact post-transplantation improvement and tissue recovery [31,57,58]. The number of cells and timing of transplantation are also crucial factors determining the outcome of neural progenitor cell transplantation [79]. Just as the status of CNS organoids in vitro depends on the fabrication process and growth environment, the optimal age and size of organoids for transplantation can vary based on the types of disease. Different pathological processes must be examined to determine suitable organoid conditions for transplantation. For example, TBI and stroke have distinct disease progressions, inflammation types, and post-damage physiologies and recovery processes [80]. Furthermore, studies on age-dependent post-transplantation improvement have primarily focused on cerebral organoids, which differ significantly from organoids for other CNS regions such as midbrain, cerebellum, and spinal cord. The appropriate conditions for organoids in these region-specific contexts remain to be elucidated.

In addition to cellular features, the physical properties (e.g., shear strength) of CNS regions vary, necessitating consideration of different microenvironments for transplantations in different areas. Using scaffolds with comparable physical properties is critical, as it is well known that stem cell differentiation can be influenced by the physical characteristics of their surroundings [81]. Besides the physical properties of the scaffolds, the molecules they carry are critical in determining the fate of neural progenitor cells, suggesting that diverse scaffolds can be designed to fine-tune the traits of the CNS organoids [82]. These are extremely critical for successful vascularization, which majority of organoids still fail.

Including scaffolds, adequately grown patient-derived organoids provide opportunities to develop personalized therapeutics for CNS [83]. Although patients share the same disease and symptoms, the origin and pathology can largely be different depending on their genetics. Using organoid models, researchers and clinicians can obtain hints on what a patient actually suffers by understanding molecular mechanism at genetic levels. This is especially important for CNS disease as biopsy for CNS is extremely limited. At the same time, the genetic variety of the patient-derived organoid can pose a problem during organoid development. Depending on the genetics of patients, the favored microenvironments will be different which can cause unexpected problems during fabrication process [84,85]. This can also lead to low quality and batch dependent characteristics of samples. Efficient handling methods for patient-derived organoids is required to be developed.

Lastly, clinical details regarding organoid transplantation require comprehensive study. In the case of SCI, inflammatory reactions involve fluctuations in various cell types over time [86]. This variability applies to other CNS diseases and differs by region. The adaptation and therapeutic potential of CNS organoids, influenced by the microenvironments of engraftment sites, can fluctuate depending on the timing of transplantation. These effects can also be significantly influenced by recipient factors such as age and sex, which relate to the activity level of neural progenitor cells [87,88]. Technically, there remains much to research in order to develop CNS organoid transplantation clinically. However, meticulously addressing these details will make actual applications possible and enable patient-specific organoid transplantation in the future.

## Figures and Tables

**Figure 1 ijms-25-08540-f001:**
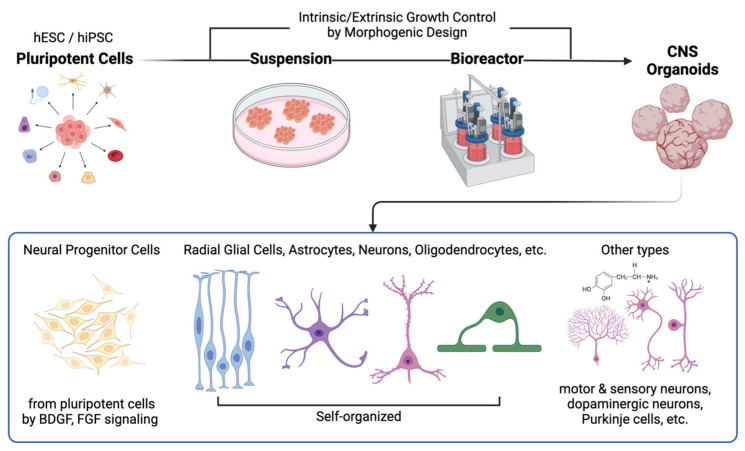
Schematic representation of the fabrication process and effects of direct implantation of human CNS organoids. CNS organoids generally originate from human embryonic stem cells (hESCs) and induced pluripotent stem cells (iPSCs) reprogrammed from somatic cells. These cells are then transferred to extracellular matrix (ECM) mimics (e.g., Matrigel) for suspension and subsequently placed in bioreactors, where the microenvironments can be controlled to promote region-specific differentiation. Depending on the morphogenetic design strategy, various cell types can be observed in CNS organoids. This process enables the creation of replicas of the human brain or spinal cord, which can be used for therapeutic applications.

**Figure 2 ijms-25-08540-f002:**
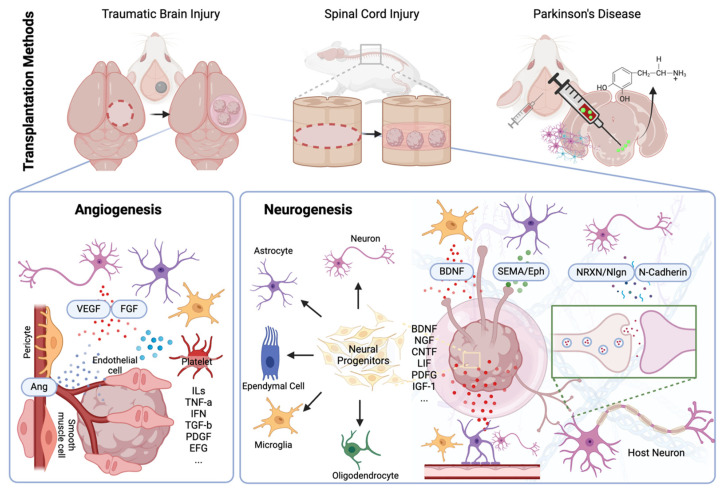
Schematic representation for CNS organoid transplantation methods. The schematic representation details the methods and mechanisms of CNS organoid transplantation for traumatic brain injury (TBI), spinal cord injury (SCI), and Parkinson’s disease (PD) using rodent models, outlining the expected postoperative therapeutic effects at the engraftment site. For TBI and SCI, direct organoid implantation is employed, while stereotactic surgery is performed for PD. Postoperative therapeutic effects are categorized into two main areas: angiogenesis and neurogenesis. At the transplantation site, inflammatory reactions occur, inducing active angiogenesis toward the transplanted organoids through factors such as vascular endothelial growth factor (VEGF), fibroblast growth factor (FGF), and angiopoietin (Ang). The transplanted organoids adapt to the host CNS via neurogenesis, aided by various growth factors from host cells. These growth factors induce neural differentiation, axonal growth (facilitated by brain-derived neurotrophic factor (BDNF), semaphorin (SEMA), and ephrin (Eph)), and synaptogenesis (by neurexin (NRXN), neurolignin (Nlgn), and N-cadherin), transforming organoids into functional tissue integrated with the host matrix.

**Table 1 ijms-25-08540-t001:** Comparison of investigated parameters: organoid versus conventional stem-cell therapy (neural progenitor cells; NPCs) [32,50,51,52].

Parameters	Degree of Effects		Description
	NPC	Organoid	
Transplanted cell survival	↑	↑↑↑	A higher number of transplanted cells remaining for organoids at both 2 and 4 weeks
Transplanted cell apoptosis	↑↑	↑	Transplanted cell apoptosis notable in NPC grafts.
Host immune response	-	-	No serious immune response for both cases
Angiogenesis	↑	↑↑↑	− Significant detection of an endothelial cell marker (CD31) in organoid grafts − Host-originated vasculature detected both in peripheral areas and the graft center in organoid grafts − Shorter vasculature in both organoid and NPC grafts − Disorganized donor cells related to vasculature in NPC grafts
Neural proliferation	↑↑	↑↑↑	− Higher donor cell proliferation density in organoid grafts − Cells organized at ventricular side in organoid grafts − Disorganized proliferating cells in NPC grafts − Comparable number of neural progenitors in both graft types
Neural differentiation	↑	↑↑↑	− More prominent differentiation of neurons, astrocytes, intermediate progenitors, and deep layer neurons in organoid grafts − Notable colocalization of neuronal differentiation with host cells in organoid grafts − Comparable differentiation of oligodendrocytes in both graft types
Axonal growth	-	↑↑	− Neurofilament heavy chain (NF-H) detected in organoid grafts with long projections − No NF-H signal detected in NPC grafts

**Table 2 ijms-25-08540-t002:** Recently conducted studies on CNS organoid transplantation.

Transplantation Purpose	Source of Organoid	Type of Organoid	Achievements	
Adaptation of grafted organoids	hESC	Cerebral organoid	Observation of axonal growth, vascularization, and neural activity in engrafted organoids	[25]
Comparison of neural stem cell and cerebral organoid transplantation	hESC	Cerebral organoid	Superior neurogenesis observed in cerebral organoids compared to neural stem cell transplantation	[32]
Repair of TBI	hESC	Cerebral organoid	Effect of transplanted organoid age on adaptation and growth of engrafted cells in organoids	[31] [57]
	hESC and hiPSC	Sheared cerebral organoid	Extended subcortical projection and electrophysiological maturity of grafted organoids	[58]
	hESC	Cerebral organoid	Cortex tissue recovery and improvements in cognitive function	[78]
	hESC	Cerebral organoid	Reduction of GFAP expression and promotion of neural repair by improving glial scar	[53]
	hiPSC	Cortical organoid	Observation of in vivo specific features of cortical organoids and identification of their effects on animal behavior	[21]
Repair of SCI	hESC	Cerebral organoid	Promotion of axon regeneration and neural network and recovery of motor function	[63]
	Human astrocytes	Spinal cord organoid from directly reprogrammed neuroectodermal cells	Spinal cord-specific neuronal growth from spinal cord organoid and its synaptic connection with host neurons	[49]
	hESC	Spinal cord motor neuron organoid with decellularized neonatal spinal cord matrix (DNSCM)	In vivo maturation of the grafted organoid advocated by signaling molecules from DNSCM	[23]
	human spinal cord-derived neural cells	Vascularized spinal-cord-like tissue	In vitro vascularization of spinal cord organoid with linear orientation promoted neural regeneration	[34]
Dopamine level recovery of Parkinson’s disease model	hESC and hiPSC	Midbrain-like neural stem cells (isolated from midbrain organoids in vitro)	Reversal of motor function by increased dopamine release from transplanted dopaminergic neurons	[43]
	hiPSC	Midbrain organoid	Dopaminergic cell differentiation of midbrain organoid in vitro and reversal of motor function by increased dopamine release in vivo	[22]

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
