# Peer review of "Therapeutic Transplantation of Human Central Nervous System Organoids for Neural Reconstruction"

_ijms, 2024, doi:10.3390/ijms25158540_

Round 1
Reviewer 1 Report
Comments and Suggestions for Authors
In this review, the authors highlight the fact that many treatments for damage to the central nervous system are not available today. Many of these damages can lead to irreversible neurological deficiencies. The review highlights the potential role of organoids in direct repair at injury sites. In particular, the authors critically analyze and evaluate preclinical transplantation results, advantages and disadvantages. The use of organoids could be used in the future to study the phenomenon of neuro-regeneration.
The review is well written and clear; the topic is of great interest to the scientific community, especially of great interest for the study of neurodegenerative diseases.
The figures are clear and very understandable. The literature references are complete and relevant.
Section 3.3 Parkinson's Disease lacks a brief reference to the hallmarks of the disease: the role of synuclein (aggregation) and neuromelanin.
The review deserves publication after the suggested addition of a small reference to the hallmarks of Parkinson's disease
Reviewer 2 Report
Comments and Suggestions for Authors
Please see attached document

Comments on the Quality of English LanguageMinor editing of English language required
